# At the Origins of Tobacco-Smoking and Tea Consumption in a Virgin Population (Yakutia, 1650–1900 A.D.): Comparison of Pharmacological, Histological, Economic and Cultural Data

**DOI:** 10.3390/biology10121271

**Published:** 2021-12-03

**Authors:** Matthias Macé, Camille Richeval, Ameline Alcouffe, Liubomira Romanova, Patrice Gérard, Sylvie Duchesne, Catherine Cannet, Irina Boyarskikh, Annie Géraut, Vincent Zvénigorosky, Darya Nikolaeva, Charles Stepanoff, Delphine Allorge, Michele Debrenne, Norbert Telmon, Bertrand Ludes, Anatoly Alexeev, Jean-Michel Gaulier, Eric Crubézy

**Affiliations:** 1International Research Laboratory “Coevolution between Human and Environment in Eastern Siberia”, 37 Allées Jules Guesde, 31000 Toulouse, France; matthias.mace@orange.fr (M.M.); patrice.gerard@univ-tlse3.fr (P.G.); sylvie.duchesne@inrap.fr (S.D.); dariyasakha@gmail.com (D.N.); bertrand.ludes@u-paris.fr (B.L.); alekan46@mail.ru (A.A.); 2Pôle Biologie Pathologie Génétique, Unité Fonctionnelle de Toxicologie, CHU Lille, CHU, Bd Pr. J. Leclerc-CS 70001, 59037 Lille, France; camille.richeval@chru-lille.fr (C.R.); delphine.allorge@chru-lille.fr (D.A.); 3ULR 4483-IMPECS-IMPact de l’Environnement Chimique sur la Santé, Faculté de Médecine-Pôle Recherche, Université de Lille, 1 Place de Verdun, F-59045 Lille, France; 4UMR 5288, Centre for Anthropobiology and Genomics of Toulouse, CNRS, Université Toulouse III, 37 Allées Jules Guesde, 31000 Toulouse, France; ameline.alcouffe@univ-tlse3.fr (A.A.); liubomira.romanova@univ-tlse3.fr (L.R.); norbert.telmon@univ-tlse3.fr (N.T.); 5Institute of Foreign Languages and Regional Studies, North-Eastern Federal University, 56 Belinskogo Street, 677000 Yakutsk, Russia; 6Institute of Forensic Medicine, 11 Rue Humann, 67085 Strasbourg, France; catherine.cannet@wanadoo.fr (C.C.); annie.geraut@iml-ulp.u-strasbg.fr (A.G.); 7Central Siberian Botanical Garden SO RAS, 630090 Novosibirsk, Russia; irina_2302@mail.ru; 8UMR 2029, CNRS, University of Paris Descartes, 2 Place Mazas, 75012 Paris, France; zvenigorosky@unistra.fr; 9Laboratoire d’Anthropologie Sociale, 52 Rue du Cardinal Lemoine, 75005 Paris, France; charles.stepanoff@ehess.fr; 10Department of Romano-Germanic Philology, Institute of Humanities Novosibirsk State University, 1 Pirogov Roas, 630090 Novossibirsk, Russia; m.debrenne@g.nsu.ru; 11Institute for Humanities Research and Indigenous Studies of the North (IHRISN), Petrovskogo Street 1, 677007 Yakutsk, Russia

**Keywords:** herbal teas, green tea, theobromine, theophylline, caffeine, nicotine, cotinine, mummies histology, pipe

## Abstract

**Simple Summary:**

We want to study (i) how tobacco and tea spread in a population with no previous contact with these substances and (ii) what factors influenced their diffusions. We measured the rates of theobromine, theophylline, caffeine, nicotine, and cotinine in hair samples from 47 frozen bodies from eastern Siberia which date from the contact with Europeans to the assimilation of people into Russian society (end of 19th century). We compared our results with archaeological and historical data. The factors that influenced the spread of smoking could be family factors, the proximity of sale outlets, and fashion. Long before tobacco, herbal teas existed, and the generalization of tea seems to have happened in the 19th century under pressure from economic circuits and tea ceremonies diffused by the elites, even though the habit of herbal tea drinking certainly persisted. Some epidemiological characteristics of present-day Yakutia could find their origins in the 18th and 19th-century diffusion phenomena.

**Abstract:**

(1) Background: The way tobacco and tea spread among virgin populations is of major interest our understanding of how ancient economic and cultural practices could have influenced current habits. (2) Methods: hair concentrations of theobromine, theophylline, caffeine, nicotine, and cotinine were measured in hair samples from 47 frozen bodies of people from eastern Siberia, dated from the contact with Europeans to the assimilation of people into Russian society. (3) Results: hair concentration of theobromine, theophylline, and caffeine vary with the type of beverage consumed: green, black, or local herbal teas. Shortly after the first contacts, a few heavy consumers of tobacco were found among light or passive consumers. Tobacco-related co-morbidities began to be recorded one century after and heavy tea users were only found from the 19th century (4) Conclusions: Economic factors and social and family contacts seem to have played a decisive role in tobacco consumption very early on. Behavioral evolution governed the process of substance integration into Siberian culture and was a determinant for the continuity of its use across long periods of time. Analyzing the respective contributions of social and economic processes in the use of these substances opens avenues of investigation for today’s public health.

## 1. Introduction

The use of tobacco products is the deadliest addiction worldwide [1], and researchers have found a positive association between smoking and caffeine intake in some populations. However, the preference for the type of caffeinated drink—mainly tea versus coffee—varies significantly among countries [2]. Despite prevention campaign efforts to reduce the public health burden of tobacco, prevention has not benefitted all strata of various societies [3] because smoking and caffeine intake involve the interplay of pharmacology and genetics as well as social factors and environmental factors, including fashion and identity [4]. Although multilevel transdisciplinary models are needed to understand sources of tobacco-related health disparities comprehensively, the incorporation of historical context into tobacco-related health disparities research remains infrequent. Yakutia (northeastern Siberia) offers a unique opportunity for such research due to the preservation of xenobiotics related to tobacco and tea consumption in the frozen bodies of autochthonous people [5]. Furthermore, the current population of the Sakha Republic exhibits generalized tea-drinking habits and a higher incidence of smoking than any other part of the Russian Federation [6,7], which allows for historical comparisons. To this end (Figure 1), we investigated the levels of the main compounds related to tea and tobacco consumption in hair sampled from 47 frozen bodies, dating from the earliest contact with Europeans (1620 AD) to the assimilation of native people into Russian society (end of the 19th century). For some of them, we carried out pulmonary histological investigations for potential co-morbidities. We compared these analyses with cultural data, such as the presence of pipes in the tombs and even that of a teapot of English origin, the oldest imported teapot found in the north. This work aims to understand how tobacco and tea spread among virgin populations as a historical basis to provide new insights into current tea and tobacco consumption.

## 2. Materials and Methods

### 2.1. General Context

Yakuts represent the largest ethnic group of the Sakha Republic of Eastern Siberia (Yakutia), with almost half a million individuals. Although their history remains largely unknown, they descend from a population that migrated northwards from the Lake Baikal area, following the expansion of the Great Mongolian Empire in the 13th century, and the present-day Yakuts are the descendants of these ancient Yakuts [8]. Except for the Arctic and Antarctic, Yakutia is the coldest country in the world, with winter temperatures dropping below −70 °C and annual temperature amplitudes over 100 °C. People and cattle adapted to extreme climatic conditions, and they have developed a traditional lifestyle based on livestock herding, where horses, cattle and reindeer provided means of transport and sources of meat, milk, and clothing [9]. The first contacts of the autochthonous population with the Russians date back to 1620 AD, but there was almost no trace of these contacts in material culture before 1689/1700 AD [10]. Trappers and itinerant merchants made the first contact, but then trade developed. The inhabitants of Yakutia traded with Russia and arrivals from the East as well as China via the trading posts of Nertchinsk and then of Kiakhta, south on the Chinese and Mongolian borders. From the end of the 18th century, the ports of Okhotsk and Ayan on the Pacific coast became more and more critical for imports [5]. Samples recovered from burials are not always random because, before the 19th century, most individuals were not buried. In the 17th century, only male clan chiefs were buried, whereas after 1700 AD, as many men as women were buried. They were mainly from the elite, and the assumed identity of several subjects and/or their status could be inferred from the associated artefacts, modes of burial, and sometimes local legends that specified who they are (chiefs, shamans) [5]. Assimilation in the Russian Orthodox culture was fully accomplished after 1800 AD. Russian colonization had a far-reaching impact on the social sphere, including the rise of an economic elite in the early 18th century [8], the reduction of seasonal mobility and the settlement in villages, and the later progressive decline of earlier shamanic religious practices in favor of Christianity [5]. The exposure of the immunologically-naive Yakut population to Russian germs also resulted in massive epidemiological outbreaks of smallpox [10], tuberculosis [11] and some changes in cultural practices and diet that affected dental health [12]. According to our historical research, it does not appear that cholera affected Yakutia.

### 2.2. Data Collection

We have excavated more than 150 frozen bodies in four geographical regions (31,000 km²) of the Sakha Republic (Yakutia) (Figure 1) [13]. We classified them into five chronological periods: before 1700, 1700–1750, 1750–1800, 1800–1850, and after 1850 AD [14]. We categorized and analyzed the characteristics of these ancient subjects and studied their familial relationships, social status, gender, and age [15]. There are some multiple graves with relatives who died within a short time of one another which will allow us to examine whether there are intra-family variations. Among these, 47 had well-preserved head hair not in contact with the sediment. The sample was genetically homogeneous [8], and all subjects had the same hair color (black).

Hair samples were collected and stored during the autopsy of the subjects following procedures ruling out the possibility of outside contamination. In the laboratory, hair samples (*n* = 47) were photographed and decontaminated in 2 × 2 min baths in water, followed by 2 × 1 min baths in dichloromethane. After drying proximal hair segments (2 to 3 cm length), we cut them into small pieces (<1 mm length). We analyzed proximal hair segments (2 to 3 cm in length) close to the skull corresponding to the last few weeks of life. It is challenging to know if these few cm represent individual consumption accurately because, at certain times, there could be supply difficulties during part of the year. One element of the discussion will therefore concern the relationship between the analyses and other archaeological elements to specify to what extent our results can be considered representative.

### 2.3. Hair Analysis

Concentrations of the main compounds related to tea and tobacco consumption were analysed. In one case, we found remains of tea leaves in a teapot associated with a woman, and we analysed them (Figure 2). For extraction, we incubated 20 mg of hair samples in methanol, and we used the solution obtained after centrifugation for analysis. Previous studies on nicotine and caffeine analyses in hair and the use of liquid chromatography with high resolution mass spectrometry detection (LC-HRMS) or with high tandem mass spectrometry detection (LC-MS/MS) tools for toxicological investigations in hair [16,17] supported our decision to use these two analytical methods to detect tobacco and tea products in our ancient population sample. We combined these two techniques according to previously published methods [18,19,20] screening for drugs and toxic compounds, psychoactive drugs including natural alkaloids, and tea methylxanthines (caffeine, theobromine, theophylline) and tobacco alkaloids (nicotine) and metabolites (cotinine).

1—A large screening for drugs and toxics compounds was performed in hair samples using LC-HRMS. We used a Waters ultra-performance liquid chromatography system consisting of two binary solvent manager LC pumps, a sample manager autosampler, and an Acquity column manager oven. Mass spectrometry data were acquired using a XEVO G2-XS QTOF instrument (Waters, Manchester, UK) controlled with MassLynx 4.1 software. After the addition of 50 µL of internal standards (methyl-clonazepam and β-OH-ethyltheophyllin) to 100 µL of the methanolic hair extract, the obtained mixture was subsequently evaporated to dryness at +30 °C under a gentle stream of nitrogen. The dry residue was reconstituted using 50 µL of a mixture of ammonium formate buffer 5 mM, pH 3/acetonitrile in 1% formic acid (87/13; *v*/*v*): 10 µL were injected in the chromatographic system. Chromatographic separation was performed using an ACQUITY HSS C18 column (150 × 2.1 mm, 1.8 µm, Waters) in an oven at 50 °C, and mobile phases including ammonium formate buffer 5 mM at pH 3 and acetonitrile in 0.1% formic acid. For detection, mass spectrometric conditions were as follows: positive electrospray ionization interface (ESI+), ion spray voltage set at 20 V, source temperature set at 140 °C and desolvation temperature at 500 °C with a desolvation gas flow rate of 900 L/h, nitrogen as desolvation gas, and argon as collision gas. The data were processed using ChromaLynx, TargetLynx, MassFragment, and MetaboLynx associated software (Waters) using a homemade database of more than 1400 substances.

2—Using LC-MS/MS (XEVO TQ-S system, Waters, Manchester, UK), another screening for psychoactive drugs including natural alkaloid compounds and related metabolites was performed in hair samples. After the addition of 50 µL of internal standards (methyl-clonazepam and β-OH-ethyltheophyllin) to 50 µL of the methanolic hair extract, the obtained mixture was subsequently evaporated to dryness at +30 °C under a gentle stream of nitrogen. The dry residue was reconstituted using 100 µL of ammonium formate buffer; 5 mM at pH 3: 10 µL was injected for separation using an Acquity UPLC HSS C18 column (150 × 2.1 mm, 1.8 µm, Waters) and a gradient including ammonium formate buffer 5 mM at pH 3 and acetonitrile in 0.1% formic acid. A Xevo TQ-S tandem mass spectrometer was used for detection after positive electrospray ionization mode in the MRM mode (using two transitions for each analyte) for more than 300 compounds and metabolites.

3—Lastly, the detection in hair of caffeine, theobromine, theophylline, nicotine, and cotinine was performed using the same LC-MS/MS device (XEVO TQ-S system, Waters, Manchester, UK). Calibration curves were extemporaneously achieved in the 0.1 to 20 ng/mg range for caffeine, theobromine, and theophylline, and in the 0.2 to 20 ng/mg range for nicotine and cotinine using a blank head hair sample. After the addition of 50 µL of internal standards (nicotine-D4 and cotinine-D3 at 5 µg/L) to 100 µL of the methanolic hair extract, the obtained mixture was subsequently evaporated to dryness at +30 °C under a gentle stream of nitrogen. The dry residue was reconstituted using 100 µL of ammonium formate buffer 5 mM at pH 3. Chromatographic separation of this extract was performed using an Acquity UPLC HSS C18 column (150 × 2.1 mm, 1.8 µm, Waters) and a gradient of (A) ammonium formate buffer 5 mM at pH 3 and (B) ACN/0.1% formic acid as mobile phase at a flow-rate of 0.4 mL/min during 5 min. The oven temperature was set at 50 °C and the injection volume was 10 µL. The gradient elution started with 100% of A and decreased to 50% at 3 min, and to 5% at 3.3 min. The washing step with 95% of solution B was held from 3.3 min to 4 min and the initial condition was applied from 4.1 min to 5 min. A Xevo TQ-S tandem mass spectrometer was used for detection after positive electrospray ionization mode in the MRM mode using the following transitions: *m*/*z* 195.2 to 138.0 (for quantitation) and *m*/*z* 195.2 to 110.0 for caffeine; *m*/*z* 181.1 to 68.9 (for quantitation) and *m*/*z* 181.1 to 95.9 for theobromine; *m*/*z* 181.1 to 124.0 (for quantitation) and *m*/*z* 181.1 to 68.9 for theophylline; *m*/*z* 163.2 to 132.0 (for quantitation) and *m*/*z* 163.2 to 117.0 for nicotine; *m*/*z* 177.1 to 79.9 (for quantitation) and *m*/*z* 177.1 to 98.0 for cotinine; *m*/*z* 167.2 to 136.0 (for quantitation) and *m*/*z* 167.2 to 134.1 for nicotine-D4; and *m*/*z* 180.1 to 80.0 (for quantitation) and *m*/*z* 180.1 to 100.9 for cotinine-D3. The validation procedure applied for this analytical method complied with both the French Analytical Toxicology Society (SFTA) and international recommendations for the validation of new analytical methods [21,22] including linearity, limit of detection (LOD), lower limit of quantification (LLOQ), accuracy, precision, and matrix effects. Calibration curves, estimated using 1/x weighted quadratic regression, were considered acceptable if the coefficient of determination (r^2^) was at least 0.99. The LLOQ was the lowest concentration with the two transitions’ presence and an intra-assay precision CV% and a relative bias lower than 25%. Intra-day and inter-day precision were calculated by analysing 3 concentration levels in five replicates on five different days. Relative standard deviation (RSD) and percentage deviation of the average concentration from the corresponding nominal value were used to estimate precision and accuracy, respectively. These parameters were considered acceptable when they were lower than 25% at the LLOQ, and lower than 20% for other levels. Ion suppression phenomenon was studied following the experimental system previously proposed [23]. Briefly, a standard solution containing the compounds of interest (at 100 µg/L) was continuously and directly infused into the mass spectrometer interface. A simultaneous LC flow containing either a pure mobile phase or a blank biological extract (blank hair) was introduced through a tee. Evolution of the signal of the transitions at the retention times of the corresponding compounds of interest was studied to determine the presence and intensity of ion suppression.

Chemical and reagents β-OH-ethyltheophyllin, methyl-clonazepam, 5-sulfosalicylic acid, ammonium formate, and formic acid, were purchased from Sigma-Aldrich (Saint-Quentin-Fallavier, France). LC-MS grade water and acetonitrile were purchased from Biosolve (Dieuze, France), while acetonitrile and methanol HPLC grade and 30% hydrochloric acid were purchased from VWR Prolabo (Fontenay-sous-Bois, France).

### 2.4. Lungs and Left Breast Sample Preparation

From our sample of 150 bodies, we were able to study lung histology from 14 subjects. In 7 out of these 14 cases, we checked hair and lungs concomitantly to establish whether high nicotine use was associated with anthracosis. For one woman (shamanic tree, #2), we endeavoured to ascertain if she smoked while breastfeeding because she presented high cotinine and nicotine levels, and one of the children—deceased between one and three years—that we excavated in the same grave was her son [15] (whose hair was not preserved).

Lungs were rehydrated in Ruffer I solution for 3 h and then immersed in 10% neutral buffered formalin for 7 days. After fixation, tissues were dehydrated through increasing grades of ethylic alcohol, cleared in xylene and embedded in paraffin wax. Tissue sections of 5 μm thickness were stained with hematoxylin and eosin (H&E) to assess the general morphology.

Left breast: Tissues were naturally mummified by freezing and remained in this state until they were removed from the funeral chamber and autopsied. Upon reception in the laboratory tissues were rehydrated, given their very dry aspect, before the histological processing. Tissues were rehydrated in Ruffer I solution. Rehydration was completed after 3 h.

Once rehydrated, tissues were fixed in 10% buffered formalin for 4 days. One third of the tissues was deep frozen and the remaining pieces dehydrated through increasing grades of ethylic alcohol, cleared in xylene, and embedded in paraffin wax. Serial sections of 4 μm in thickness were cut and stained with: (i) Hematoxylin and Eosin (H&E) to assess the general morphology on paraffin sections, (ii) Giemsa stain for the demonstration of parasites on paraffin sections, and (iii) Sudan III and IV for the demonstration of lipids on frozen sections.

### 2.5. Relationship between Methylxanthines and Substances

To compare our results with historical data, we conducted an ethnopharmacological and historical study to determine what substances could have been consumed by the Yakuts at different periods. We then searched the literature in English and Russian for the composition of the various substances that could be the natural source of the alkaloid compounds of interest in this study. Our search strategies used a combination of standardized terms related to vegetal material (e.g., plant, tree, fruit, flowers) or the exact name of the vegetal, and keywords that we implemented in (i) NCBI PubMed (1900–present) and Google Scholar (1900–present); (ii) the data library of the N.N. Vorozhtsov Novosibirsk Institute of Organic Chemistry; and (iii) old monographs and articles of Belarus where alkaloids were very much studied but not available on the Internet. We also consider publications that were not found in the literature search but cited in retrieved publications. Unfortunately, the absence of tests on living organisms or significant bibliographic data concerning the relationships between the substances ingested and hair patterns of methylxanthines will oblige us to present these data in the form of hypotheses that we will compare with historical data.

### 2.6. Statistical Analyses

Principal Component Analysis was computed over the levels of the five pooled hair analytes, and outliers were defined qualitatively on the PCA projection.

#### 2.6.1. Logistic Regression for Hair Concentration throughout Time

Multivariate logistic-regression analysis was used to assess a putative shift in the use of tobacco and tea with time. The dependent variable was binary coded: 0 for individuals dated before 1800 and 1 for individuals dated after 1800. The independent variables were the hair concentrations in the five studied xenobiotics and potential confounding factors that were adjusted for in the multivariable analyses included sex, age (0 to 14, 15 to 29, 30 to 49, and >50), social status (suspected shaman or not), and presence of an associated pipe (absence, presence: simple one, special one, and imported one). Computations were made on data including and excluding outliers.

#### 2.6.2. Association between the Number of Metabolites and Time

The number of tea components detected in hair varies from 0 to 3 methylxanthines (theophylline, theobromine, caffeine) depending on the subject. We tested the variations over time (before and after 1800) for three possible combinations: theobromine alone versus all others, theobromine + caffeine versus theobromine + caffeine + theophylline, and all three components versus nothing.

#### 2.6.3. Correlations Nicotine/Cotinine among Age Classes throughout the Time Range

Correlation by age class: Spearman correlation coefficient was computed between nicotine and cotinine concentrations within each age class (0–15, 15–30, 30–50, and over 50 years old).

Co-correlation between age classes: Co-correlation between age classes was assessed between the children age class (0–15 y/o) and each one of the others. The computation was performed with data including and not including the 4 outliers (#14, 29, 31, 41). A one-sided Fischer Z test was conducted with the null hypothesis being that correlation is lesser in the child group than in adult group and the alternative being that data for the children display a lesser correlation coefficient. Zou’s confidence intervals [24] were also computed. Computations were made using the Cocor R package [25].

#### 2.6.4. Geographic Variation of Xenobiotic Hair Concentrations and Pipes

The association between xenobiotic hair concentrations and geography was assessed using several permutations-based methods. First, a permutation ANOVA was performed using the four archeological fields (North, East, West, and South) as groups [26]. We then assessed correlation (Spearman’s rank test) between analyte concentrations and distances from the three main trading posts and/or fairs in the east through which the product was likely to transit and then the known entry areas (North-East and East) through which it was likely to arrive if it had been shipped to one of the ports on the Pacific coast (Okhotsk or Ayan) or the west, south-east and north-west, routes of arrival in Yakutia of products from the southern trading posts (on the Mongolian or Chinese border, Kiakhta or Nerchinsk) and/or from Russia. Strength of association was tested using a permutation over individual concentration.

As the associations between the levels of cotinine and/or nicotine with the pipes are statistically significant (9/10), we implemented the same methods of association on the totality of the pipes (for a majority of subjects the hair was not preserved), before and after 1800 AD. We present only the one before 1800 AD (20 pipes), the only positive one.

#### 2.6.5. Interpolation & Mapping of PCA by Product

Tea and tobacco use were then approximated by combining the correspondent chemical xenobiotics by PCA (methylxanthines for tea, nicotine and cotinine for tobacco). Resulting PCA first axis values were interpolated over a rhomboid area comprised between the four studied archaeological fields using spline interpolation and triangulation based on Renkas tripack [27] for linear interpolation as implemented in the “akima” R package. The interpolated values were overlaid according to their geographic coordinates over the map as black transparency (values normalized between 0 and 0.7). Maps were produced using the “marmap” R package and the National Oceanic and Atmospheric Administration bathymetrics and global relief data (resolution of 5 min) and then drawn following an orthographic projection for 100° longitude, 60° latitude, and 0 m elevation.

## 3. Results

### 3.1. Tea and Tobacco Related Components Detected in Hair

Hair analysis showed an LOD of 0.01 ng/mg and LLOQ of 0.02 ng/mg for nicotine and cotinine and a LOD of 0.05 ng/mg and LLOQ of 0.01 ng/mg for caffeine, theobromine, and theophylline (Table 1). We did not detect other drugs or toxic compounds (including natural alkaloid compounds) in the 47 hair samples. The second dichloromethane bath tested negative for all compounds and all samples.

Above, LOD is the limit of detection and LLOQ is the lower limit of quantification. The fit of the 1/x weighted quadratic regression (analyte-to-IS peak area ratio versus theoretical concentration) was verified for all the compounds. The coefficients of determination were superior to 0.99 for each calibration curve (from LLOQ to 20 ng/mg). Precision and accuracy were acceptable for all substances, meeting the criteria set: lower than 25% at LOQ and lower than 20% at other levels for both inter- and intra-day assays. The samples are listed in chronological order, from 1600 AD to 1900 AD. We note the increase of subjects presenting the three methylxanthines (caffeine, theobromine, and theophylline) after 1800 AD (generalization of the black tea, dark grey), the presence of subjects with two compounds (theobromine, caffeine) between 1700 and 1800 AD (emergence of the green tea, medium grey), and the presence from 1600 AD of only theobromine (local tea: Ivan tea, light grey).

Theobromine (*n* = 31) was more often detected (including in one child, 6 to 9-month-old) than caffeine (*n* = 29), which was itself more often detected than theophylline (*n* = 19). After 1800 AD, all subjects had at least one substance, and 8/15 had all three. The subject presenting the highest concentration was a woman buried with a teapot. The remains of the tea leaf from the teapot tested positive for caffeine (+++), theophylline (++), and theobromine (+), whereas her hair tested positive for these three substances but with a concentration of theobromine higher than that of theophylline. Before 1800 AD, in 8 subjects, no substance was detected, and only 6/32 had all three; the most frequent case was that of subjects presenting both theobromine and caffeine (9/32), essentially women (8/9: Fisher’s exact test *p* = 0.0033 for comparison between genders). The only male who was tested positive was from the period prior to 1700 AD. There are some multiple graves containing relatives who died within a short time of one another. Two women of the same family (numbers 2 and 6) and a woman and one child (numbers 10 and 11), buried together, show little or no trace of components from tea and tobacco. Two women (numbers 26 and 12), mother and daughter [15] who died of smallpox at the same time [10], have different levels of tea and tobacco components; the brother of the daughter (#29) shows no presence of tea components. Brother and sister were heavy smokers (numbers 12 and 29), and she was breastfeeding at the time of death. Some substances could be associated with special social status (shaman), as evidenced by a sub-significant association between the status and theobromine concentrations (Table 2). No significant association was evidenced for gender differences. However, a trend could be confirmed by a greater sample given the much lesser *p*-values observed after 1800 AD.

We compare correlation (co-correlation) between age classes in Table 3 and Table 4. It shows an increase in the correlation with age. The difference was significant between the young (0–15 years old) and the oldest quartile (over 50 years old).

From the end of the 18th century, an older woman was the only case with the level of cotinine associated with the highest level of nicotine; she died with anthracosis pigments associated with emphysema (Figure 3).

Eighty-two per cent (39/47) of the subjects had traces of tea and tobacco. The highest users of the two substances were from the 19th century, and we only found high tea users in the 19th century. Moderate tea users in the 19th century were also tobacco users.

### 3.2. Changes in Drinking Substances and Tobacco Use over Time

For individuals living before 1800 AD, caffeine hair concentration ranged between 11 ng/mg and 252 ng/mg (median 28 ng/mg), while for individuals living after 1800 AD, it ranged between 10 and 9304 (median 55) (Table 5)—there is a significant sub-association (*p*-value = 0.0562). We do not find a significant association for the other xenobiotics (Table 6). Before 1800 AD, some subjects had no tea components; after 1800 AD, all had at least one, and a majority had all three (*p* = 0.0264). From 1700 to 1750 AD, most individuals had theobromine and caffeine; their number decreased drastically from 1750 to 1800 AD, and there were no more after 1800 AD (*p* = 0.0182). There were no differences between individuals with only theobromine at any time (*p* = 1).

### 3.3. Xenobiotic Hair Concentrations and Pipes over Geographical Range

As for previous analyses, we considered two periods: before and after 1800 AD. For individuals dated before 1800 AD, we do not find a significant association between geographical regions and the xenobiotic hair concentration. After 1800 AD, in the case of caffeine and cotinine, we were able to find a correlation between the distance from sampling location to the three principal trading posts and/or fairs in the east through which the imported goods had to transit. The highest correlation was with the northeast, which represents the entry point into Yakutia of the trade route by which merchants brought tea at this period from the Pacific harbour of Okhotsk (Table 7
*p*-value = 0.0112). The robustness of this association was challenged by permuting 10% of the xenobiotic hair concentration values among individuals (10,000 replicates). We confirmed an association with a probability of observing rs above 0.5 above 0.9 for Zachiversk (Table 8). This observation led us to suspect a NE-SW gradient in caffeine concentration. We do not find a significant association between geographic data and any of the other xenobiotics. Substance drinking and tobacco use interpolated over the entire range between the principal places of Yakutia are shown in Figure 4. Before 1800 AD, only isolated individuals showed high levels of derived compounds in hair, and after 1800 AD, levels showed a geographic gradient for tea (Figure 5), but a gradient was not detectable for tobacco. For pipes, before 1800 AD, we were able to find a correlation between distances, from the location of graves containing pipes and to the city of Yakutsk (where goods and merchants necessarily transited) (Table 8 and Figure 6).

For Caffeine and Cotinine, *p* is the probability of observing rs above 0.5 among replicates. For Pipes, *p* is the probability of observing an association (*p*-value < 0.05) among replicates.

Before 1800 AD, no geographical pattern was found as only isolated individuals showed significant levels of methyxanthines associated with tea. After 1800 AD, a significant negative correlation was observed between caffeine concentrations and the settlements situated northeast (above 80% significant correlations observed after 10.000 permutations tests) on the path of the importations from Okhotsk harbor (Table 8).

### 3.4. Histological Analysis

Analysis of the lungs: Eight lungs out of 14 presented traces of anthracosis (Figure 3). There was no relationship with gender (five women, three men), or with age (15-year-old subjects presented traces; some over 60 did not). Two subjects with very low levels of nicotine did not show signs of anthracosis, while in two other subjects, nicotine levels and presence/absence of anthracosis were in opposite directions. Bakhtahh 1, an older woman who was found with a unique model of pipe, was the only case with a high level of cotinine associated with the highest level of nicotine; she died with anthracosis pigments associated with emphysema (Figure 7). 

### 3.5. Histology of the Left Breast

The abundant alveolar aspect of the gland and the scarcity of the collagen fibers are suggestive of a lactating or a pregnant breast (Figure 8).

## 4. Discussion

Our study is the first to analyse many hair samples collected in the minutes or hours following the excavation of frozen bodies. Considering that the growth rate of human scalp hair ranges from 0.7 to 1.4 cm/month with an average of about 1 cm/month [28] we analysed samples of hair corresponding to approximately two to three months’ growth before the subject’s death, and much less for the children for whom the hair was not very long.

Classically, researchers find that xenobiotics are well preserved in hair [29], and several associations suggest the robustness of our results: (i) The ten subjects buried with a pipe had traces and/or significant hair concentrations of cotinine and/or nicotine, and one had no trace of nicotine and traces of cotinine. However, in this last case, it was an imported European earthenware pipe—the only one known in Yakutia—which indicates that the pipe was more of a prestige object than an object for smoking (Figure 7). (ii) The nicotine/cotinine correlation increased with age and the correlation was significantly different between the young (0–15 years old) and the oldest quartile (over 50 years old). One of the two subjects with the lowest cotinine level was a child aged 6 to 12 months, and for four subjects where cotinine was detected alone, these levels were low. Present-day studies provide subjects with low levels of cotinine not associated with nicotine [30]. (iii) An older woman from the end of the 18th century was the only subject with a level of nicotine associated with the highest cotinine level; she died with anthracosis pigments associated with emphysema. (iv) In graves dated before 1800 AD, we found a sub-significant association between the presence of a pipe and the distance to Yakutsk or the distance to the southwest of the studied area.

The measurement of nicotine hair concentration can be an informative tool to assess tobacco smoke exposure (whether active or passive) and related illnesses. The main pitfall of nicotine hair concentration interpretation is the impossibility to discriminate between active and passive exposure [31,32,33]. Indeed, a positive result for nicotine in hair (usually greater than 0.2 ng/mg) could not confirm with any certainty an active use of tobacco by the deceased during their lifetime. Nicotine hair contamination from external sources (such as smoking by other members of the communities in which the deceased individuals lived) cannot be ruled out. Several authors reported the inability of this method to differentiate between active and passive exposure [31,32,33]. In fact, observed hair concentrations for smokers range usually between 0.4 and 11 ng/mg and between 0.9 and 11 ng/mg for nicotine and cotinine, respectively. However, comparable hair values are reported in non-smoker populations, especially in the case of passive exposure [34]. All in all, the determination of nicotine (and/or cotinine) in mummy hair samples cannot definitively confirm active nicotine use during the lifetime of the deceased, as external contamination and exposure to second-hand smoke during their lifetime cannot be excluded [35].

Evolution over time of tobacco smoking: The only individual of the 17th century (Djoussulen) presented only a few traces of cotinine and nicotine in hair, perhaps reflecting passive exposure due to contact with Europeans as some of his grave goods demonstrate. Less than a century after contact, proximity to the major tobacco import trading post of Kiakhta was a determining factor in consumption, as were direct relationships of individuals with this spot. The development of economic circuits affected consumption, which largely flouted laws—as early as 1634 AD, smoking was prohibited in Russia on punishment of death (1634), and it remained so for autochthonous people until 1822 AD [36]. The development of smoking was linked to the economic circuits in the 18th century, while in the 19th century tobacco consumption became generalized and economic circuits were detectable only for tea which was imported via the Pacific harbors. Between 1700 AD and 1800 AD, two men of the elite (#29 and #30) with a Mongolian haircut and a southern fashion for their clothes were heavy consumers of tobacco. One of them was found with a beautiful mammoth ivory pipe (Figure 7). The manufacturing of highly attractive smoking accessories was instrumental in the dissemination of tobacco. The custom then was to pass the pipe from one person to the other to smoke thus encouraging family transmission. These men probably travelled to a trading post on the Mongolian border in contact with China where tobacco, tea, and furs were traded [5]. Two other contemporary heavy smokers, females in this case (#12 and #26), were related to one of them and one was breastfeeding when she died. The mother of #29 and #30, who was not supposed to be living under the same roof as them when she died, was not a heavy tobacco user and drank undoubtedly a different tea than her daughter (see below). Similar family habits were also observed in the 19th century.

Currently, in most individuals, tea and caffeine are present at very high concentrations compared to the two other components [37]. Theophylline and theobromine hair concentrations range between 0.1 and 0.6 ng/mg and between 0.3 and 10 ng/mg, respectively [37]. In case of a positive result for caffeine, theophylline, or theobromine in hair (usually greater than 0.1 ng/mg), there is no data in the literature allowing us to assess the level of tea drinking by the subject (even if our sample the teapot was associated with the highest tea consumer). Tea contains some 3-methylxanthines (caffeine, theophylline, and theobromine), but its chemical composition is significantly affected by tea processing [38], leaf maturation, botanical variety, geographical origin, agricultural practice [39], and preparation [40]. The influence of physiological variations from one subject to another on the incorporation of these methylxanthines in hair has not, to the best of our knowledge, been described. While all three 3-methylxanthines are present in most black teas, theophylline is often absent in green teas [41] and some black teas [42] (Table 9). Following these observations, we could hypothesise that black tea consumption, which appeared punctually before 1750 AD, became widespread in the 19th century, which is consistent with historical data [43].

The type of drink has been assigned to each subject according to the levels of theobromine, theophylline, and caffeine measured in the hair. Herbal tea is associated with the presence of theobromine only. Green tea was assigned when both theobromine and caffeine were present. Coffee is assigned to subjects with caffeine only. Black tea is assigned to subjects with all three xenobiotics associated with tea consumption (theobromine, caffeine, theophylline).

The level of tobacco consumption has been defined by taking into account the average of the two compounds associated with tobacco (nicotine and cotinine), the orders of magnitude of the two measures being too different to analyse them separately. A distribution curve of these averages has made it possible to define four groups of consumption: low consumer (nicotine and cotinine under 200 ng/mg), medium consumer (between 200 and 501 ng/mg), high consumer (between 501 and 2500 ng/mg), and very high consumer (over 2500 ng/mg).

In addition to these imported substances, we identified 19 other local plants that may have been drunk or smoked during historical times in Yakutia. Only a few contain the methylxanthines involved. An herbal tea based on (or including) the roots of *Rhodiola rosea*, which contains caffeine in small quantities [44], is known in the Siberian people’s pharmacopoeia for its anti-fatigue, anti-ageing effect. Three senior women had low levels of caffeine, and we excavated two of them in Villuy, one of the only places in Yakutia where it grows [45]. Ivan tea, a fermented herbal tea from *Chamerion latifolium* and *Chamerion angustifolium*, which was drunk at that time, contains only theobromine in large quantities [45]. It could have been consumed by five subjects of various ages from the 17th and 18th centuries; in the 19th century, we only suspected Ivan tea in three subjects, two of whom were less than one year old (Table 9). Two subjects had only traces of theophylline; three senior women had only traces of caffeine, and three had traces of both caffeine and theophylline. This association could be related to herbal teas that we could not identify or passive exposure to tea. In the summer, Yakuts boil water before drinking it, and the container used to boil the water may have been used as a teapot beforehand.

It is highly challenging to give hypotheses about the substances’ origins because several substances could be drunk or mixed. Thus, two individuals from the 18th-century with a high level of theobromine compared to theophylline and caffeine levels in their hair were able to consume green tea and/or black tea and at the same time as Ivan tea. For example, the remains of tea leaf from the teapot tested positive for caffeine (+++), theophylline (++), and theobromine (+), whereas the hair of the woman buried with the teapot tested positive for these three substances but with a concentration of theobromine higher than that of theophylline. The tea preparation for the funeral meal may have been different from that consumed during the last months of the deceased’s life.

All periods combined, we found a few heavy tobacco consumers, men and women, amongst light or passive consumers, but tobacco-related co-morbidity was only recorded after 1750 AD (in a woman) at a time when a large part of the population, even older children, became addicted according to travelers’ accounts [46,47]. Heavy tea users were only detected in the 19th century and the heaviest users of the two substances dated from this century. In the 19th century, while the Yakut population assimilated into the Russian orthodox way of life, several contemporary trends were detected: all subjects seemed to become at least light consumers of tea, and moderate tea users were also tobacco users. The generalization of tea and tobacco is linked to economic circuits (smoking in the 18th century and tea in the 19th century). In our study area, tobacco use spread faster than tea use, probably because tobacco is intrinsically more addictive than tea, and because of the absence of a previous tradition in the use of these substances. A variety of herbal teas—especially Ivan tea—have been used by the Yakuts for centuries and it is not impossible that we have detected it. The use of green tea by women in the 18th century also supports this hypothesis, and the generalization of the use of tea follows the development of tuberculosis [11] for which green tea was used as a treatment, perhaps with some results [48]. In the 19th century, tea had made the transition from a beverage used in medicine and spirituality-based socialization (shamans?) to being used by elites with the tea ceremony led by women of fashion and means. The ceremony dictating that tea must be brewed and drunk using specialized equipment soon became established as a more generalized practice [44]. For tobacco, there were no previous traditions; socio-economic and family contacts seem to have played a decisive role very early with this substance, which is more addictive than tea. In contemporary populations, numerous epidemiologic studies have consistently demonstrated positive associations between the use of tobacco, alcohol, and caffeine [49], and it would be interesting to determine whether the same association existed among the Yakuts. To address this question, we have only historical records. The popularity of alcohol seems to have been delayed until the early 19th century [47,50]. It was banned from sale to the local population in 1822 [51]. However, similar to tobacco before it, it continued to be illicitly traded, and the authors note the increasing consumption of tea and alcohol in the 19th century [52,53].

## 5. Conclusions

In a virgin population in the 18th century, factors that influenced the spread of smoking could be the same as those that today are often highlighted as having a role in tobacco consumption and described, due to their interrelationship, as ‘rhizomatic smoking’ [54]: family factors [55,56], the proximity of sale outlets [57,58], and fashion [59,60]. In this Yakut sample, we found passive smoking among children and maternal smoking and breastfeeding [61]. However, only 8 lungs out of 14 presented traces of anthracosis, which could be a reasonably low percentage in subjects from the past [62] due to the inhalation of smoke from solid fuel—in particular, biomass (wood) and smoke that, in populations from the past, came from enclosed rooms inside their dwelling places [63]. In the 19th century, the state of health of the population could have deteriorated [12]; we should note that drinking tea at high temperatures could be associated with an increased risk for esophageal cancer when combined with excessive alcohol or tobacco use [64]. Long before tobacco, herbal teas existed, and the generalization of tea seems to have happened later. In the 19th century, its generalization occurred under pressure from economic channels and tea ceremonies diffused by the elites, even though the habit of herbal tea drinking certainly persisted. The current population of the Sakha Republic exhibits generalized tea drinking habits; a higher smoking incidence than any other part of the Russian Federation [6]; and an actual prevalence of smokers in the north [7]. Some epidemiological characteristics of present-day Yakutia could find their origins, in part anyways, in the 18th and 19th-century diffusion phenomena supported by the findings reported in this study. Analyzing the respective contributions of social and economic processes in using these substances opens avenues of investigation for today’s public health.

## Figures and Tables

**Figure 1 biology-10-01271-f001:**
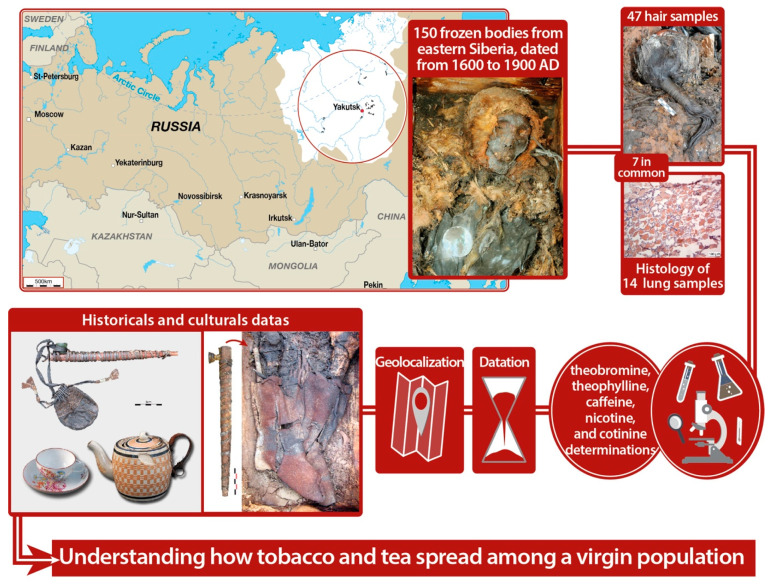
Graphical abstract of the methodology with a map of Yakutia.

**Figure 2 biology-10-01271-f002:**
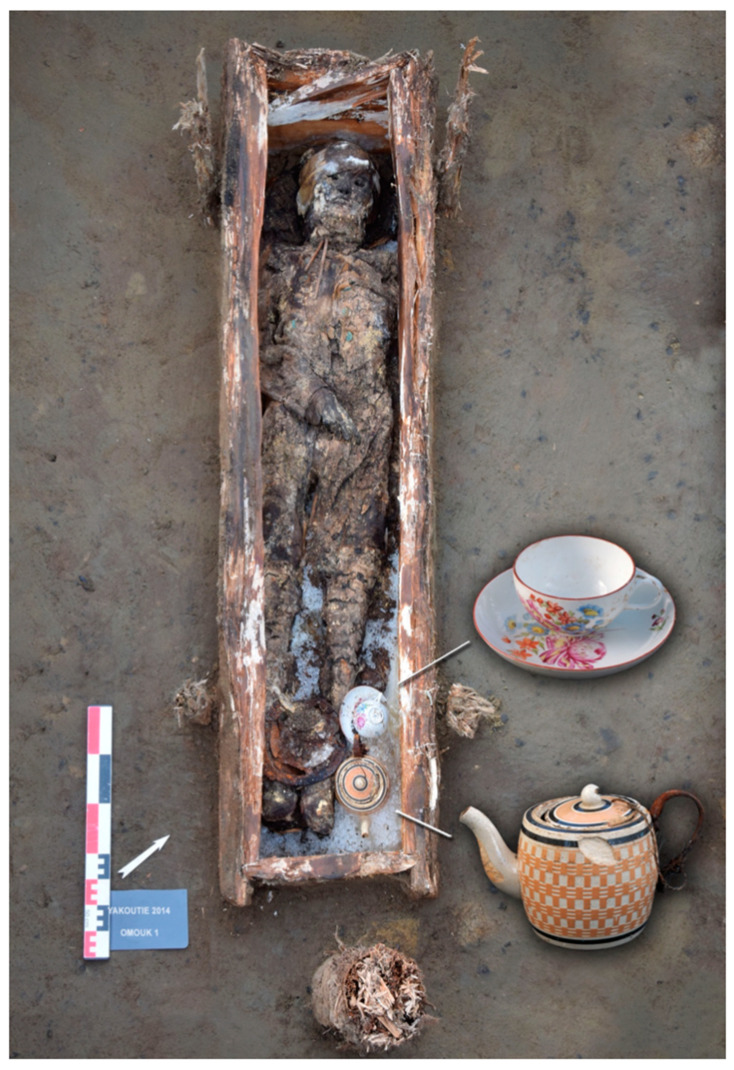
Omouk 1 grave with artefacts: teapot and cup, and under the legs and feet the smoking kit (iron lighter, fire bag, pipe). This grave (around 1820 AD,) was found on the Indigirka River, Arctic Circle. The cup originated from the Russian factory A. Popov known for its solid porcelain. The teapot is of English origin. The woman’s hair had high levels of nicotine, cotinine, and the highest concentration of caffeine, theobromine, and theophylline.

**Figure 3 biology-10-01271-f003:**
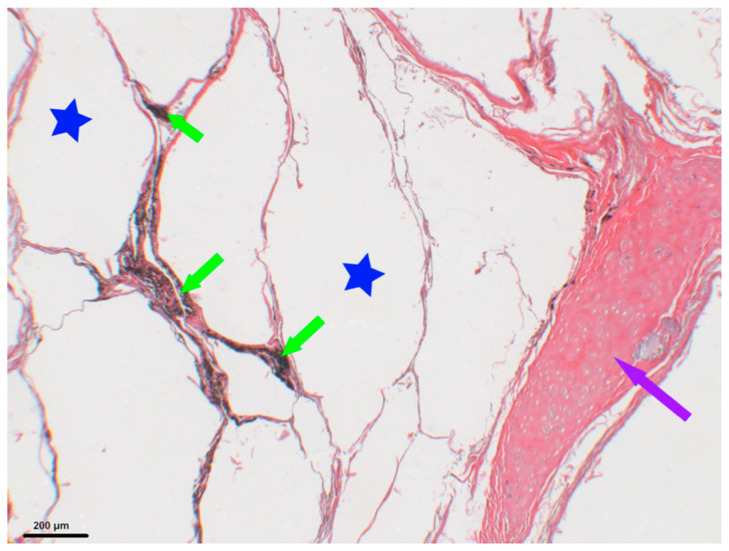
Histological slice of Baktakh_1 right lung (old woman). Anthracosis is shown by green arrows, associated emphysema by blue stars and bronchus hyaline cartilage by the purple arrow.

**Figure 4 biology-10-01271-f004:**
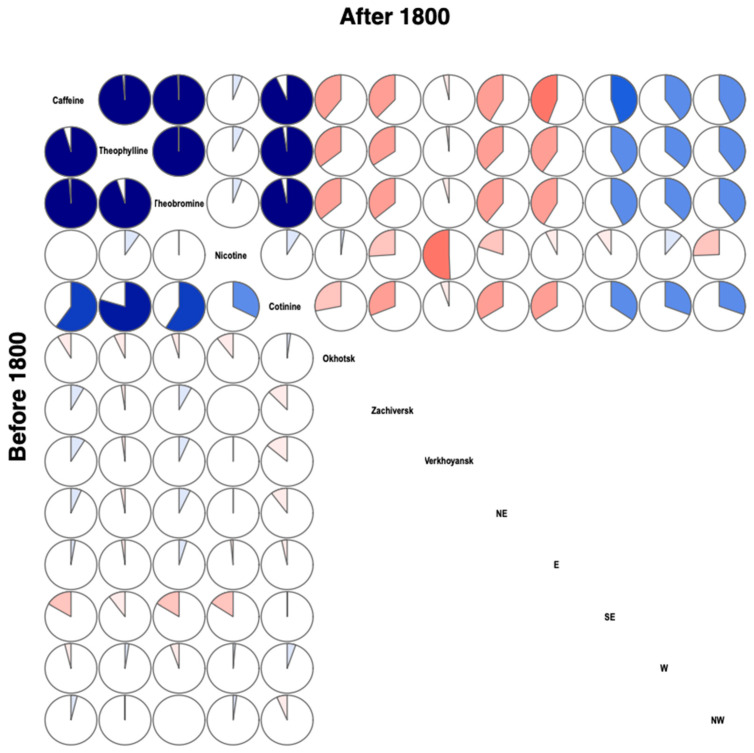
Correlogram displaying biochemical/geographical correlations. Upper-diagonal part: before 1800. Lower-diagonal part: after 1800. Positive correlations are shown in blue, negative in red. Color intensity shows absolute strength in correlation. Negative correlations in the east and North mean that the further away from these points (distance increases), the more the concentration decreases, resulting in a gradient: dark in the NE > light in the SW.

**Figure 5 biology-10-01271-f005:**
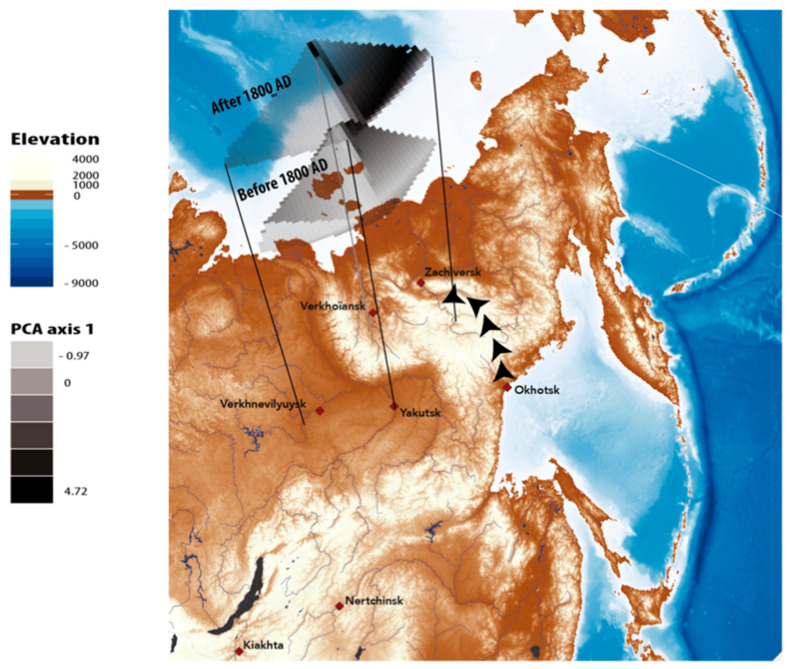
Map of Eastern Siberia displaying geographical interpolation of tea use overlaid with geographical data of the studied area. Tea use was estimated as the first axis of the Principal Component Analysis over the three studied methylxanthines (Caffeine, Theobromine, Theophylline). Eigenvalues—axis 1: 2.19; axis 2: 0.51; axis 3: 0.28.

**Figure 6 biology-10-01271-f006:**
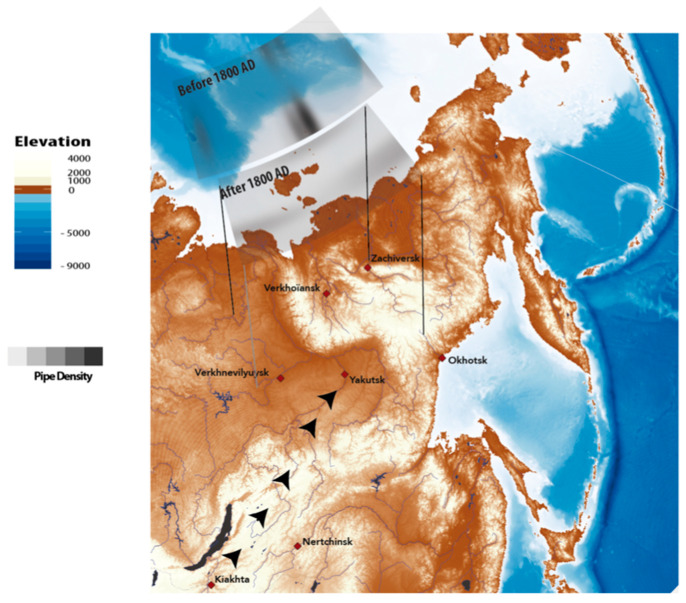
Map of eastern Siberia displaying geographical interpolation of pipes found in the graves associated to the bodies with geographical data of the studied area. Before 1800 AD there was a sub-association between pipe ownership and distances to Yakutsk and the South-West of the studied area. The south-west was the route of entry for tobacco from the Kiakhta trading post. After 1800 AD, no geographical pattern was found.

**Figure 7 biology-10-01271-f007:**
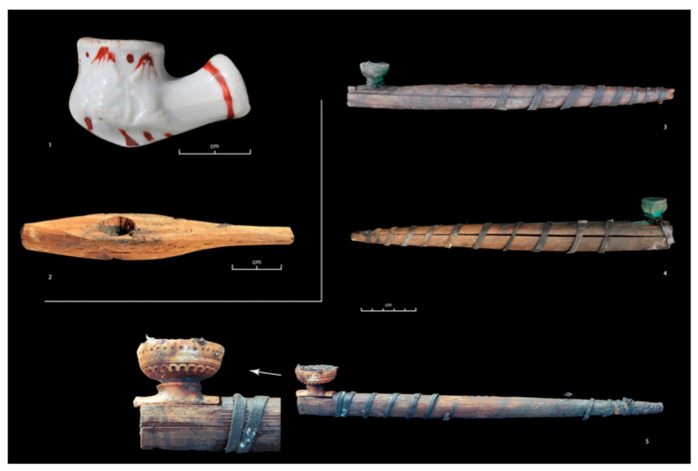
Pipes. (**1**) Pipe bowl of European origin in white porcelain—single case. Grave of a female subject aged 25 to 30 years (Oyogosse tumula 2) without any trace of cotinine or nicotine in the hair. The bowl, found without a pipe, was perhaps more an ostentatious artefact linked to the spread of tobacco fashion in the early 19th century than a useful object for smoking. (**2**) Wooden pipe of a unique design that evokes a chillum. It was found tied to a silk scarf placed in the left hand of Bakhtakh 1 (#41 a woman from the second half of the 18th century). This woman had the highest hair levels of nicotine and cotinine in the sample and suffered from pulmonary emphysema. One instance of chillum smoking leads to maximal increase in eCO levels indicating the possibility of chillum being the most dangerous mode of smoking. (**3** et **4**) Common pipe models. The pipe, made of wood, is kept closed by a leather lace and the bowl, made of copper alloy, is an imported good, resulting from trade and exchange with Europeans. These smoking accessories, produced in large quantities and easily accessible, increased tobacco consumption while meeting the demand. (3) Pipe found in the multiple tombs of Shamanic Tree 1 (Central Yakutia region, first half of the 18th century) slipped into the right boot of one of the two female subjects, the eldest (30 to 55 years old) and the mother of the other members buried in this tomb (her daughter, her son, and two grandsons, children of her daughter). (4) Pipe found in the grave of a teenager aged between 15 and 18, at the Kuranakh site (Verkhoyansk region, second half of the 18th century), accompanied by an iron lighter. (**5**) Mammoth ivory pipe, finely carved of native manufacture, found in the multiple grave of Shamanic Tree 1 (cf. No. 3) associated with a man between 30 and 50 years old. He was a heavy smoker, belonging to the elite who was in contact with the Kiakhta trade post where the exchange of tea and tobacco took place. This type of pipe, of native manufacture and carried by the elite, was an attractive smoking accessory which instrumented the dissemination of tobacco among the less privileged who bought manufactured pipe bowls (3 and 4).

**Figure 8 biology-10-01271-f008:**
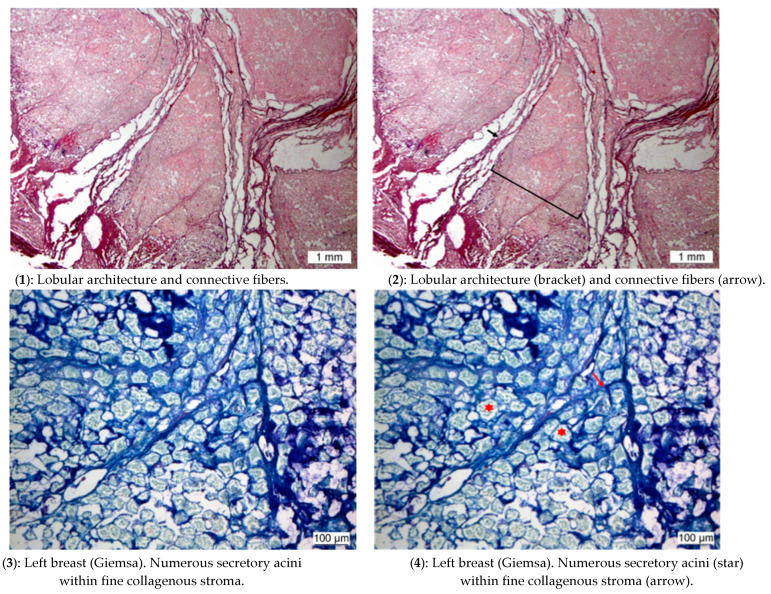
Histology of the left breast: the lobular aspect was clearly identifiable, and the lobules were separated by connective fibers (**1**,**2**). Numerous secretory acini were found within a fine collagenous stroma (**3**,**4**). The positive Sudan III and IV staining confirmed the presence of lipids within the acini (**5**,**6**).

**Table 1 biology-10-01271-t001:** Hair levels of xenobiotics (pg/mg) by individual alongside the available datations, age class, and archæological data.

*#Sample*	*Name*	*Datation AD*	*Sex*	*Age Class*	*Pipe*	*Shaman*	*Anthracosis*	*Theobromine*	*Caffeine*	*Theophylline*	*Nicotine*	*Cotinine*
*4*	Djoussulen	1600–1700	M	?				32	ND	ND	17	13
*46*	Kureleekh 2	1700–1750	F	>50			Yes	ND	11	13	171	35
*37*	Urun myran	1700–1750	M	30–50				ND	ND	ND	15	24
*10*	Tysarastaakh 2A	1700–1750	F	30–50				ND	ND	ND	13	31
*3*	Sergueleakh	1700–1750	F	15–30				ND	ND	ND	68	39
*38*	Ordiogone 1	1700–1750	M	>50				ND	ND	ND	28	67
*13*	Mounour urekh 1	1700–1750	M	>50				ND	ND	ND	126	195
*29*	Arbre 1 Sujet 1	1700–1750	M	30–50	Yes			ND	ND	ND	194	12055
*45*	Haras	1700–1750	F	0–15		Yes		388	33	16	47	22
*36*	Alyy	1700–1750	M	?	Yes		Yes	270	61	112	42	173
*26*	Arbre 1 Sujet3	1700–1750	F	30–50				219	48	42	89	89
*27*	Arbre 3	1700–1750	F	30–50	Yes			124	16	ND	124	612
*8*	Kyys	1700–1750	F	15–30		Yes		123	26	ND	801	201
*12*	Arbre chamanique femme Sujet 2	1700–1750	F	15–30			Yes	103	55	ND	1196	702
*44*	Eletcheï 2	1700–1750	F	0–15		Yes		84	ND	ND	92	92
*11*	Tysarastaakh 2E	1700–1750	M	0–15				38	51	19	ND	58
*20*	Atyyr meïte 1	1700–1750	M	15–30				37	ND	ND	19	42
*17*	Ebuguey 1	1700–1750	F	30–50				11	12	ND	20	10
*40*	Boulgounniakh 2	1700–1750	F	30–50		Yes		10	12	ND	46	40
*21*	KouraNDkh 1	1750–1800	M	15–30	Yes			ND	14	27	84	728
*16*	Sobolokh 1	1750–1800	F	30–50				ND	76	ND	32	15
*30*	Ordiogone 2	1750–1800	M	?			Yes	ND	ND	60	210	2083
*22*	Uettekh	1750–1800	F	0–15				ND	ND	ND	37	57
*24*	Mounour urekh 1 bis	1750–1800	F	>50	Yes			ND	ND	ND	127	82
*43*	Sordonokh 1	1750–1800	F	15–30		Yes	Yes	387	19	ND	58	146
*1*	Eletcheï 1	1750–1800	F	30–50			Yes	244	ND	ND	33	312
*35*	Kerdugen 1	1750–1800	M	15–30				228	252	57	66	313
*19*	Batta tchara	1750–1800	M	>50				207	15	ND	31	80
*41*	Bakhtakh 1	1750–1800	F	>50	Yes (particular)		Yes + Emphysema	104	93	ND	628	14749
*34*	Tomtor 1	1750–1800	M	30–50				50	19	11	37	106
*23*	Arbre 2 (okht2)	1750–1800	F	>50				27	ND	ND	99	82
*28*	Bouogaryma 2	1750–1800	F	>50				17	30	ND	20	194
*47*	Ottokh 1	1800–1850	F	>50				ND	10	ND	249	126
*39*	Ottokh 2	1800–1850	F	>50				ND	15	ND	88	433
*14*	Omouk 1	1800–1850	F	30–50	Yes			4712	9304	1929	214	4528
*9*	Byljasyk 1	1800–1850	M	0–15				495	791	56	125	388
*15*	Omouk 3	1800–1850	F	15–30				285	692	116	72	293
*42*	Otchugouï 1	1800–1850	F	30–50				82	ND	ND	66	100
*32*	Tomtor 2	1800–1850	M	15–30				75	55	19	ND	46
*7*	Ous siré 1	1800–1850	M	15–30	Yes			72	14	12	811	153
*25*	TargaND 1	1800–1850	Indet	0–15				49	ND	ND	ND	10
*31*	BalagaNDkh 3	1850–1900	M	30–50				ND	17	10	21	46
*2*	Oyogosse tumula 2 Sujet A	1850–1900	F	15–30				ND	ND	29	14	39
*33*	Touora urekh 1	1850–1900	F	30–50	Yes			1716	4464	607	192	562
*18*	Ken ébé 3	1850–1900	M	0–15				323	689	76	145	52
*5*	Tchenkeloeil	1850–1900	Indet	0–15				251	ND	ND	25	28
*6*	Oyogosse tumula 2 Sujet B	1850–1900	F	15–30	Yes (importation)			13	26	18	ND	22

**Table 2 biology-10-01271-t002:** Tea/tobacco use and social status (suspected shamans).

	Estimate	Std. Error	*z* Value	Pr (>|z|)
(Intercept)	−2.121772	1.161690	−1.826	0.0678
Theobromine	0.014519	0.008582	1.692	0.0907
Theophylline	−0.039655	0.047568	−0.834	0.4045
Caffeine	−0.007103	0.006212	−1.143	0.2528
Nicotine	0.002124	0.002538	0.837	0.4026
Cotinine	−0.004708	0.005562	−0.846	0.3973

**Table 3 biology-10-01271-t003:** Correlation coefficients between Nicotine and Cotinine hair levels among age class.

	0–15 y/o	15–30 y/o	30–50 y/o	>50 y/o
With outliers	0.5555	0.4529	0.4529	0.9183
Without outliers	0.5555	0.4529	0.7316	−0.1124

**Table 4 biology-10-01271-t004:** Co-correlation between children and the three other age classes.

	Comparison	N1 / N2	Fischer’s Z	*p*-Value	Zou’s C.I.
With outliers	*0–15* vs. *15–30*	8 / 12	−0.2711	0.7863	−1.0277 ; 0.6904
	*0–15* vs. *30–50*	8 / 14	−0.9274	0.3537	−1.2142 ; 0.3326
	*0–15* vs. *over 50*	8 / 10	−2.1887	0.0286	−1.4432 ; −0.0443
Without outliers	*0–15* vs. *15–30*	8 / 12	−0.2711	0.7863	−1.0277 ; 0.6904
	*0–15* vs. *30–50*	8 / 11	−1.1145	0.2651	−1.2769 ; 0.2884
	*0–15* vs. *over 50*	8 / 9	0.6765	0.4987	−0.6761 ; 1.2136

**Table 5 biology-10-01271-t005:** Changes in drinking substances and tobacco use over time.

		Estimate	Std. Error	*z* Value	Pr (>|z|)
Before 1800	(Intercept)	−0.0627	0.816549	−0.077	0.939
	Theobromine	−0.0004	0.004493	−0.082	0.935
	Theophylline	−0.0065	0.017675	−0.370	0.712
	Caffeine	0.0022	0.002603	0.833	0.405
	Nicotine	−0.0052	0.006963	−0.745	0.456
	Cotinine	0.0003	0.000664	0.397	0.691
After 1800	(Intercept)	−8.1698	7.229920	−1.130	0.258
	Theobromine	0.0769	0.062876	1.223	0.221
	Theophylline	0.0092	0.040560	0.227	0.821
	Caffeine	−0.0276	0.024864	−1.110	0.267
	Nicotine	0.0310	0.029524	1.050	0.294
	Cotinine	−0.0283	0.025701	−1.102	0.271

**Table 6 biology-10-01271-t006:** Historical evolution of xenobiotic concentrations.

	Estimate	Std. Error	*z* Value	Pr (>|z|)
(Intercept)	−0.291535	0.608502	−0.479	0.6319
Theophylline	−0.038243	0.023974	−1.595	0.1107
Theobromine	−0.002581	0.005238	−0.493	0.6222
Caffeine	0.006761	0.003541	1.909	0.0562
(Intercept)	−0.7210826	0.4288329	−1.682	0.0927
Nicotine	0.0009768	0.0015033	0.65	0.5158
Cotinine	−0.0015442	0.0019076	−0.81	0.4182

**Table 7 biology-10-01271-t007:** Absolute correlations between individual xenobiotic hair levels and distance to surrounding cities and four historical gateways into Yakutia.

	Caffeine	Theophylline	Theobromine	Nicotine	Cotinine
Yakutsk	0.6307	0.9071	0.5192	0.6967	0.6705
Okhotsk	0.1252	0.7507	0.2006	0.8799	0.4307
Zachiversk	0.0112	0.1369	0.2589	0.5717	0.2794
Verkhoyansk	0.0655	0.3088	0.6500	0.1591	0.3680
Kiakhta	0.1334	0.9336	0.3118	0.8122	0.4384
Nertchinsk	0.1420	0.9336	0.3403	0.8629	0.4541
Verkhnevilyuysk	0.1797	0.7763	0.2006	0.6806	0.4080
NE (66°27′0′′ N, 143°13′12′′ E)	0.0112	0.1369	0.1509	0.5717	0.1654
E (64°10′12′′ N, 145°7′48′′ E)	0.1097	0.7763	0.1899	0.8629	0.4541
SE (63°34′12′′ N, 126°30′0′′ E)	0.3403	0.9336	0.3118	0.8290	0.5109
W (62°15′0′′ N, 116°9′36′′ E)	0.1334	0.9336	0.3118	0.8122	0.4384
NW (66°45′36′′ N, 123°22′12′′ E)	0.3701	0.7763	0.1899	0.5717	0.5621

**Table 8 biology-10-01271-t008:** Geography and Caffeine/Cotinine/Pipes: permutation tests (10.000 replicates/10% of permuted data by replicate).

	Caffeine		Cotinine		Pipes	
	r_s_	*p*	r_s_	*p*	Pval	*p*
Yakutsk	0.1636	0.219	−0.0137	0.23	0.07	0.17
Okhotsk	−0.4909	0.304	0.0410	0.219	0.60	0.00
Zachiversk	−0.7273	0.931	−0.3508	0.236	0.92	0.00
Verkhoyansk	−0.5727	0.886	−0.3964	0.244	0.98	0.00
Kiakhta	0.4818	0.302	−0.0046	0.238	0.77	0.00
Nertchinsk	0.4727	0.298	−0.0228	0.224	0.58	0.00
Verkhnevilyuysk	0.4364	0.264	0.0137	0.225	0.88	0.00
NE (66°27′0′′ N, 143°13′12′′ E)	−0.7273	0.923	−0.3508	0.225	0.76	0.00
E (64°10′12′′ N, 145°7′48′′ E)	−0.5091	0.831	0.0592	0.208	0.30	0.00
SE (63°34′12′′ N, 126°30′0′′ E)	0.3182	0.223	0.1230	0.216	0.08	0.15
W (62°15′0′′ N, 116°9′36′′ E)	0.4818	0.319	−0.0046	0.203	0.46	0.00
NW (66°45′36′′ N, 123°22′12′′ E)	0.3000	0.227	−0.0957	0.208	0.81	0.00

**Table 9 biology-10-01271-t009:** Type of drink and level of tabacco concumption according to hair levels of xenobiotics (pg/mg) by individual alongside the available datations, age class and archæological data.

#Sample	Name	Datation AD	Sex	Age Class	Pipe	Shaman	Anthracosis	Type of Drink	Level of Tobacco Consumption
4	Djoussulen	1600–1700	M	?				Herbal tea	Low consumer
46	Kureleekh 2	1700–1750	F	>50			Yes	Coffee and Theophylline	Low consumer
37	Urun myran	1700–1750	M	30–50				Nothing	Low consumer
10	Tysarastaakh 2A	1700–1750	F	30–50				Nothing	Low consumer
3	Sergueleakh	1700–1750	F	15–30				Nothing	Low consumer
38	Ordiogone 1	1700–1750	M	>50				Nothing	Low consumer
13	Mounour urekh 1	1700–1750	M	>50				Nothing	Low consumer
29	Arbre 1 Sujet 1	1700–1750	M	30–50	Yes			Nothing	Very high consumer
45	Haras	1700–1750	F	0–15		Yes		Black tea	Low consumer
36	Alyy	1700–1750	M	?	Yes		Yes	Black tea	Low consumer
26	Arbre 1 Sujet3	1700–1750	F	30–50				Black tea	Low consumer
27	Arbre 3	1700–1750	F	30–50	Yes			Green tea	Medium consumer
8	Kyys	1700–1750	F	15–30		Yes		Green tea	Medium consumer
12	Arbre chamanique femme Sujet 2	1700–1750	F	15–30			Yes	Green tea	High consumer
44	Eletcheï 2	1700–1750	F	0–15		Yes		Herbal tea	Low consumer
11	Tysarastaakh 2E	1700–1750	M	0–15				Black tea	Low consumer
20	Atyyr meïte 1	1700–1750	M	15–30				Herbal tea	Low consumer
17	Ebuguey 1	1700–1750	F	30–50				Green tea	Low consumer
40	Boulgounniakh 2	1700–1750	F	30–50		Yes		Green tea	Low consumer
21	KouraNDkh 1	1750–1800	M	15–30	Yes			Coffee and Theophylline	Medium consumer
16	Sobolokh 1	1750–1800	F	30–50				Coffee	Low consumer
30	Ordiogone 2	1750–1800	M	?			Yes	Theophylline	High consumer
22	Uettekh	1750–1800	F	0–15				Nothing	Low consumer
24	Mounour urekh 1 bis	1750–1800	F	>50	Yes			Nothing	Low consumer
43	Sordonokh 1	1750–1800	F	15–30		Yes	Yes	Green tea	Low consumer
1	Eletcheï 1	1750–1800	F	30–50			Yes	Herbal tea	Low consumer
35	Kerdugen 1	1750–1800	M	15–30				Black tea	Low consumer
19	Batta tcharaND	1750–1800	M	>50				Green tea	Low consumer
41	Bakhtakh 1	1750–1800	F	>50	Yes (particular)		Yes + Emphysema	Green tea	Very high consumer
34	Tomtor 1	1750–1800	M	30–50				Black tea	Low consumer
23	Arbre 2 (okht2)	1750–1800	F	>50				Herbal tea	Low consumer
28	Bouogaryma 2	1750–1800	F	>50				Green tea	Low consumer
47	Ottokh 1	1800–1850	F	>50				Coffee	Low consumer
39	Ottokh 2	1800–1850	F	>50				Coffee	Medium consumer
14	Omouk 1	1800–1850	F	30–50	Yes			Black tea	High consumer
9	Byljasyk 1	1800–1850	M	0–15				Black tea	Medium consumer
15	Omouk 3	1800–1850	F	15–30				Black tea	Low consumer
42	Otchugouï 1	1800–1850	F	30–50				Herbal tea	Low consumer
32	Tomtor 2	1800–1850	M	15–30				Black tea	Low consumer
7	Ous siré 1	1800–1850	M	15–30	Yes			Black tea	Medium consumer
25	TargaND 1	1800–1850	Indet	0–15				Herbal tea	Low consumer
31	BalagaNDkh 3	1850–1900	M	30–50				Coffee and Theophylline	Low consumer
2	Oyogosse tumula 2 Sujet A	1850–1900	F	15–30				Theophylline	Low consumer
33	Touora urekh 1	1850–1900	F	30–50	Yes			Black tea	Medium consumer
18	Ken ébé 3	1850–1900	M	0–15				Black tea	Low consumer
5	Tchenkeloeil	1850–1900	Indet	0–15				Herbal tea	Low consumer
6	Oyogosse tumula 2 Sujet B	1850–1900	F	15–30	Yes (importation)			Black tea	Low consumer

## Data Availability

Not applicable.

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
