# Peer review of "At the Origins of Tobacco-Smoking and Tea Consumption in a Virgin Population (Yakutia, 1650–1900 A.D.): Comparison of Pharmacological, Histological, Economic and Cultural Data"

_biology, 2021, doi:10.3390/biology10121271_

Round 1

Reviewer 1 Report

The manuscript entitled “At the origins of smoking and tea consumption in virgin people (Siberia 17th to 20th century). Impact of current populations” presents an analysis of substances presented in tobacco and certain kind of tea (nicotine, caffeine, theobromine, theophylline etc.) in order to understand the consumption habits of tobacco and tea of people from a certain part of Russia (Yakutia, East Siberia), during the 17th to 20th centuries. The analysis consists in the measurement of the mentioned substances from hair samples of 47 frozen bodies buried in the region. The condition of the samples were good enough in order to find reliable results.

A couple of comments to consider the manuscript suitable for publication:

  • A graphic proof of the samples would clarify the methodology used in the document.
  • A brief description of the Yakutia region (climate conditions, social organization etc.) is necessary in order to understand the consumption patterns of the inhabitants and to differentiate them with other regions and the current consumption patterns.

Author Response

Thank you for your comments and requests which will improve the article. We have reworked this article extensively following requests from reviewers. Regarding your specific requests, we have added Figure 1 which describes the whole process and we have added a whole paragraph on the context of Yakutia and the issues of the study.

Thank you again.

Eric Crubézy for the authors

Reviewer 2 Report

This manuscript with an attractive title reports levels of methylxanthines, nicotine and cotinine in hair samples collected from frozen bodies excavated from a couple of sites (AD 1600-1900) in Siberia. The authors inferred the analytical results in terms of consumption of different teas and smoking by referring to archaeological remains and histology of lung. The manuscript contains impressive pictures.

However, there are also several flaws in this manuscript.

  1. The manuscript, particularly Result section, is poorly organized so that the manuscript is difficult to read. Statistical results are fragmentary described in the Result section without statistical data presentation (e.g., Line 155-156); Table 1, a major result of this study, is too busy and too small without unit; description of LOD is located at a strange position; and so on. Line 205-217, there is a repetition. Moderate quality of English also contributes to unclarity of the manuscript. Quality of this manuscript is not high.
  2. Analytical methods are not described at all but only references are given. Some more description of the methods is required with a detailed description about analytical quality assurance. Pretreatment of hair samples (“decontamination” according to the authors, Line 56) should be clearly described.
  3. The authors analyzed hair samples taken from 2 to 3 cm from the scalp, thus the analytical results are expected to reflect the subject’s last few weeks. If some of the study subjects died of diseases, but not by accidents, then is it appropriate to expect the last few weeks life reflected the subjects’ usual life habit?
  4. It seems questionable to specify kind of tea and/or coffee consumed by the subjects based on patterns of methylxanthines in hair. The authors’ interpretation should be validated by proper references or by hair methylxanthine pattern of known consumers (e.g., living subject) of different kind of teas. Otherwise, the authors’ interpretation of tea consumption cannot be readily accepted.
  5. It may be true that the temporal trend of tea consumption and smoking habit deduced from hair analysis is consistent with historic facts, but the authors’ description seems too definitive and the “story” suggested in this manuscript seems too subjective.
  6. Although the authors label the smoking subjects as “addiction”, what was the criteria of hair cotinine levels for addictive smoker? Specify it with a reference.
  7. The subtitle “Impact on current population” is not appropriate. The authors should not overstate/over-interpret the study result to this extent.

Author Response

Thank you for your comments and requests that will improve the article; we agree with you. Following your and other reviewers' recommendations, we have revised the paper thoroughly and changed the title. A native speaker has proofread the article, and we hope it will be easier to read.

Regarding your specific requests:

1/ the outline has been changed. We have given a figure (figure 1) that summarizes the approach. We have reorganized the totality of the results by making what used to be in the supplementary data directly in the article.

2/ The analytical methods are now described in detail

3/ As there is no available date (I;e hir patterns of methylxanthines or hair concentration) to support any identification of kind of tea and/or coffee consumed by the subjects, we have removed these data from the results, and we present them as hypotheses in the discussion

4We thus gave the "history" more as a hypothesis than a definitive synthesis, and the contribution of new historical data allows us to put it in a more general context

5/ the term addiction has been removed and replaced by consumption.

6/ as previously stated, the title has been changed.

We hope to have answered your requests; thank you again.

Eric Crubézy for the authors

Reviewer 3 Report

The study subject is very interesting and importance both of traditional biology and public health.

The following comments are offered as constructive suggestions with the goal of improving the manuscript.

This study would be fully understood by thoroughly examining the lifestyle of the cold region of Siberia.

1)
Compered to warmer regions, citizens may be exposed to risk of cancer too often, such as drinking hot drink or smoking in a closed building.

Please discuss the caution of generalization or reproducibility in modern society.

2)
In public health view, smoking status is associated with alcohol consumption.

The discussion should mention alcohol consumption from the results of this study.

Unless specified the discussion from this study, please add what you can predict from previous research.

3)
In Russia, cholera outbreaks have been confirmed in early 20th century.
Drinking water, the main cause of cholera, is also closely related to subjects of this study.
If the pandemic in the past may affects the results of this study, please add this point.

Best regards.

Author Response

Thank for your comments. Due to the reviewers' comments,  even the title has been changed. Regarding specific remarks :

the relationship with alcohol has been taken into account thanks to historical research that involved our three historians in the last days. It seems that we now have the totality of the quotations on this issue in the historical period in Yakutia

The previous research, especially on anthracosis and mummies, has been taken into account and the risks of cancer. These data are presented in the discussion.

The only texts that mention cholera in Yakutia are administrative texts that say that cholera is spreading from Vladisvostock and should be taken care of, but it seems that it never reached Yakutia. Cholera followed the trans-Siberian route, and that it did not pass through Yakutia. We have therefore implemented that there was no cholera.

Thank again.

Round 2

Reviewer 2 Report

The manuscript has been improved.